# Adulteration Detection of Edible Bird’s Nests Using Rapid Spectroscopic Techniques Coupled with Multi-Class Discriminant Analysis

**DOI:** 10.3390/foods11162401

**Published:** 2022-08-10

**Authors:** Jing Sheng Ng, Syahidah Akmal Muhammad, Chin Hong Yong, Ainolsyakira Mohd Rodhi, Baharudin Ibrahim, Mohd Noor Hidayat Adenan, Salmah Moosa, Zainon Othman, Nazaratul Ashifa Abdullah Salim, Zawiyah Sharif, Faridah Ismail, Simon D. Kelly, Andrew Cannavan

**Affiliations:** 1Environmental Technology Division, School of Industrial Technology, Universiti Sains Malaysia, Penang 11800, Malaysia; 2Analytical Biochemistry Research Centre (ABrC), Inkubator Inovasi Universiti (I2U), Kampus SAINS@USM, Universiti Sains Malaysia, Lebuh Bukit Jambul, Bayan Lepas 11900, Penang, Malaysia; 3Faculty of Pharmacy, Universiti Malaya, Kuala Lumpur 50603, Malaysia; 4Malaysian Nuclear Agency, Kajang 43000, Bangi, Selangor, Malaysia; 5Surveillance Branch, Food Safety and Quality Division, Ministry of Health Malaysia, Presint 3, Federal Government Administrative Centre, Putrajaya 62675, Malaysia; 6Veterinary Public Health Laboratory, Department of Veterinary Services, Bandar Baru Salak Tinggi, Sepang 43900, Selangor, Malaysia; 7Food Safety and Control Subprogramme, Joint FAO/IAEA Centre of Nuclear Techniques in Food and Agriculture, Department of Nuclear Sciences and Applications, International Atomic Energy Agency, Vienna International Centre, P.O. Box 100, 1400 Vienna, Austria

**Keywords:** edible bird’s nest, mid-infrared, near-infrared, adulteration, multi-class discriminant analysis

## Abstract

Edible bird’s nests (EBNs) are vulnerable to adulteration due to their huge demand for traditional medicine and high market price. Presently, there are pressing needs to explore field-deployable rapid screening techniques to detect adulteration of EBNs. The objective of this study is to explore the feasibility of using a handheld near-infrared (VIS/SW-NIR) spectroscopic device for the determination of EBN authenticity against the benchmark performance of a benchtop mid-infrared (MIR) spectrometer. Forty-nine authentic EBNs from the different states in Malaysia and 13 different adulterants (five types) were obtained and used to simulate the adulteration of EBNs at 1, 5 and 10% adulteration by mass (a total of 15 adulterated samples). The VIS/SW-NIR and MIR spectra collated were subsequently processed, modelled and classified using multi-class discriminant analysis. The VIS/SW-NIR results showed 100% correct classification for the collagen and nutrient agar classes in authenticity classification, while for the other classes, the lowest correct classification rate was 96.3%. For MIR analysis, only the karaya gum class had 100% correct classification whilst for the other four classes, the lowest rate of correct classification was at 94.4%. In conclusion, the combination of spectroscopic analysis with chemometrics can be a powerful screening tool to detect EBN adulteration.

## 1. Introduction

In the Chinese community, edible bird’s nest (EBN) is considered a premium food that boosts the health of an individual during sickness and aging [1]. EBN is the structure used by swallows where they lay their egg and shelter their young swiftlets [2]. The bird’s nest mainly comes from three genera of swiftlet, namely *Aerodramus*, *Apus* and *Collocalia* [3]. The nest is constructed using the saliva from the male swiftlet’s sublingual salivary gland [4]. Before the emergence of house-farming, the bird’s nest was obtained from the wild, mainly from caves, after the swiftlets left their nest. However, wild EBNs are becoming rare, and production has started to be taken over by house farming, which is now popular in Malaysia to cater to the increasing demand by the Chinese consumers, thus contributing significantly to the industry. In its raw form, the bird’s nest is hardly edible, as it is contaminated with dirt, feather and grass. After processing by sorting, washing, and cleaning, the strands of the bird’s nest are unravelled, and impurities are removed. All these procedures are performed by manual labour. The strands of the bird’s nest are then placed into a mould in the shape of a cup to retain their original shape. The moulded EBN undergoes a drying and grading process before they are packed and sold to the market. EBN is usually consumed by cooking together with tonic food and sugar. EBN’s nutritional values (water-soluble protein, carbohydrates, iron, inorganic salt) [5], high price, up to 6000 USD/kg in China, which varies with grade [6], and the laborious work involved for the harvesting and processing make EBN a valuable delicacy.

Due to its high value, EBN is prone to economically motivated adulteration. Adulteration can be carried out during the cleaning or packaging process. Usually, the choice of adulterant material would be of those easily available, low in price and similar in appearance to the authentic EBNs. Some adulterants are indistinguishable from genuine EBN via visual inspection, such as white fungus, karaya gum and red seaweed [7]. Various approaches have been studied to reveal the presence of adulterants, including the combination of SYBR green polymer chain reaction with dimensional electrophoresis [5], X-ray microanalysis and SDS-PAGE electrophoresis [7], nitrite, nitrate and colour determination [8], gas chromatography [1] mass spectrometry [9], colorimetric sensor array [10], loop-mediated isothermal amplification (LAMP) [11] and FT-IR [12].

The most common chemometric techniques are principal component analysis (PCA) and multi-class discriminant analysis such as orthogonal partial least square-discriminant analysis (OPLS-DA). It is useful when a vast amount of data is needed to be analysed, such as the spectroscopic data, where it is too complex to perform the mathematics manually. Chemometrics can help to explore the correlation, variation, and classification between different samples. The different techniques can also be combined to achieve a better result.

However, a controlled environment is needed to carry out the tests, and those methods require well-trained personnel to perform the analyses. Therefore, despite all the methods available for the authentication of EBN’s, there is a pressing need to explore field-deployable rapid screening techniques with minimal sample preparation, to permit rapid decision making in the industry setting. Portable spectroscopic methods fit these criteria nicely.

Mid-range IR (MIR) has been used in the study of authentication of EBNs [13] but not with handheld near-infrared. Near-infrared (VIS/SW-NIR) spectroscopy shares similarities with mid-IR, as it is non-invasive, requires no sample preparation and is fast and sensitive. The applications of VIS/SW-NIR particularly in the food industry were mainly in quality and process control [14]. Furthermore, the emergence of affordable handheld VIS/SW-NIR makes this technique more advantageous in terms of mobility or portability. Among the examples of handheld VIS/SW-NIR which has been involved in previous works are Consumer Physics [15,16,17], Viavi [18], Thermo Fisher Scientific [19] and Spectral Engines Oy [20]. These devices are usually paired with their respective data processing software, or the data may be used in third-party software for the work of adulteration detection using chemometrics models. Mobility is an essential criterion for on-site screening purposes, which is beneficial to authorities and consumers. This greatly reduces the time and cost, as it facilitates screening of the suspected non-genuine samples before proceeding to more time-consuming and expensive confirmatory analysis such as hyphenated mass spectrometry. Furthermore, this hand-held VIS/SW-NIR is equipped with a short-wavelength sensor. Previous studies with VIS/SW-NIR had been focusing on the long-wavelength combination region, 1100–2500 nm, where strong interference from water absorption can be found. The restriction on long-wavelength VIS/SW-NIR is due to the lack of technology such as the appropriate sensor, whereas the recent development of a short wavelength sensor that covers the spectral range 700–1100 nm, enables higher penetration depth of samples and less interference from water absorption. Specifically, the specifications of VIS/SW-NIR (SCiO) are LED light source, Si photodiode (array, 12 elements) detector, undefined spectral resolution and signal-to-noise ratio. In contrast, MicroNIR Pro ES 1700 (Viavi) has a tungsten halogen light source, InGaAs (array, 128 elements) detector, wavelength range from 908–1676 nm, 1000–2000 nm spectral resolution and 23,000:1 signal-to-noise ratio [21]. VIS/SW-NIR (SCiO) has distinctive specifications compared to the other NIR instruments. This type of short wavelength (700–1100 nm) VIS/SW-NIR sensor can be applied to various food products to measure parameters such as moisture, protein, fat and sugar content [22] and had been applied in the evaluation of food components such as milk powder [23], salmon roe [24] and pork [25]. Furthermore, studies with MIR have been carried out extensively on different types of food fraud which include the adulteration of hazelnut [26], raw milk [27], onion powder [28], herbs [29] and black pepper [30]. These studies demonstrated the firm foundation of the MIR technique in detecting food adulteration. Hence, MIR is included in this study as a benchmark performance against the newly available handheld VIS/SW-NIR to detect EBN adulteration.

It is likely that some of the functional groups present in the adulterants are significantly different from authentic EBNs; hence, even at low concentrations, the spectroscopic device combined with chemometrics can detect their fraudulent addition. The objective of this research is to assess an in situ (point of use) screening method using a hand-held VIS/SW-NIR, the SCiO^TM^, to ascertain its ability to detect adulteration of EBN against the laboratory-based benchtop FT-IR as a benchmark technique.

## 2. Materials and Methods

### 2.1. Samples Preparation

A total of 77 sample (49 authentic EBNs, 15 adulterated EBNs and 13 pure adulterant samples) spectra were collected. The 49 authentic EBNs samples were collected from different locations (7 states) across Malaysia in the year 2016 as listed in Table 1. The EBNs samples were milled into powder form using a “Tefzel” (ethylene tetrafluoroethylene) container (DuPont, Wilmington, DE, USA) with a stainless grinding ball (1.4034, hardness approx. 52 HRC). The commonly used adulterants consisted of pure melamine (1 sample from Sigma-Aldrich Handels GmbH (Vienna, Austria);), karaya gum (1 sample from Sigma-Aldrich Handels GmbH (Vienna, Austria); and another sample from local store), nutrient agar (1 sample from Oxoid, Basingstoke, Hampshire, UK and another 4 samples from local store), collagen (1 sample from Lennox Health Products Sdn. Bhd., Selangor, Malaysia), and another sample from local store) and gelatine (1 sample from Lafleureshop.com, USA and another 2 samples from local store). For instance, karaya gum and nutrient agar are majoritively made up of carbohydrates [31,32], melamine has different form of non-protein amino nitrogen compounds [33], and gelatine is made of protein with water binding and emulsifier properties [34], whereas collagen is a natural protein derived from skin, tendon and bone [35]. The chemical composition of adulterants is similar but different to the EBN’s properties. Since only 5 types of adulterants can be procured from reputable chemical manufacturers, it was decided that another 8 adulterant samples should be purchased from local retailers and added to the database. The 5 types of adulterants from reputable chemical manufacturers were then mixed with pure EBNs to produce admixtures containing adulterants at the levels of 1, 5 and 10% (mass/mass). This step was performed by combining proper quantities of adulterants with authentic EBNs. For example, 1% adulteration was performed by adding 1 g of an adulterant into a selected authentic EBN so that the total mass of the sample added up to be 100 g. The admixture was then homogenised by using a hand-action shaker for 30 min. A similar procedure for the adulteration method can be found in the previous study by Adenan et al. [36].

### 2.2. Instrumentation

#### 2.2.1. The Hand-Held VIS/SW-NIR

A pocket-size molecular sensor the SCiO^TM^ (Consumer Physics, Israel) was used in this study. The sensor has an approximate dimension of 67.7 × 40.2 × 18.8 mm and weighs approximately 35 g, which enhanced the portability (Figure 1). This device has a broadband LED light source and Complementary Metal Oxide Semiconductor (CMOS) sensor. It can operate between the temperature range of 5 to 40 °C. It is a diffuse reflectance VIS/SW-NIR optical spectrometer, which operates and synchronises with a smartphone application, web-based processing software and cloud storage. The cloud serves as a processing and storage platform for the spectral scans, of which later the spectra can be pre-processed, analysed and built into different chemometrics models. The spectral data can be downloaded as a CSV file and used in other platforms, with a researcher license. Moreover, this device has advantages over the benchtop spectrometer, as it consumes low energy with a battery that lasts for hundreds of scans and requires no warm-up time. This device makes use of the near-infrared region of the electromagnetic spectrum from 740 to 1070 nm and produces a spectrum of 6 nm in resolution. It is designed for consumers’ daily usage in detecting food type, nutrition value and others. For instance, the application allows the user to differentiate raw poultry, raw beef, raw pork, chocolate, fruits and dairy products. This device is claimed by Consumer Physics to be capable of detecting the moisture content in selected crops such as corn and to determine the most suitable harvest time.

#### 2.2.2. Benchtop FT-IR

A Frontier^TM^ FT-IR spectrometer (Perkin Elmer Inc, MA, USA) was used for the acquisition of IR spectra. The FT-IR spectrometer was equipped with universal attenuated total reflectance (ATR) accessory, which is a single-reflection diamond ATR sampling module to obtain the spectra of mid-infrared. The detector for this FT-IR instrument is temperature-stabilised DTGS (deuterated triglycine sulphate). The spectrometer is controlled by Spectrum 10 software. This universal ATR accessory is designed to handle solid, liquid and powdered samples without sample preparation. The instrument provides spectra of the wavelength range from 16,667 to 2500 nm, with a resolution of 2,500,000 nm.

### 2.3. Spectra Acquisition

#### 2.3.1. Near-Infrared

The EBN powdered sample, approximately 10 g, was contained in a plastic zip-lock bag (polyethylene) with a dimension of 10 × 5 cm. The surface of the sample was flattened to give uniformity to the scanning. Then, VIS/SW-NIR spectra were acquired by scanning the sample through the sealed bag directly three times, as shown in Figure 2. The acquisition time for each scan was about 2 s, and triplicate measurements were conducted to reduce the signal-to-noise ratio and to enable the consistency of the measurement to be assessed. Full-spectrum scan reading from 740 to 1070 nm of wavelength was acquired, and all the sample’s spectra were collated for chemometric analysis. A similar scanning procedure was also applied to the adulterated EBNs and adulterant samples.

#### 2.3.2. Mid-Infrared

The mid-infrared (MIR) spectra were acquired by placing a small amount of sample (approximately one-third of a spatula) on the ATR crystal sensing surface, sufficiently covering up the ATR crystal, and a force of 115 N was applied on the sample using the built-in pressure gauge. Two spectra were collected from each sample, where the MIR spectra were collected at a spectral resolution of 4 cm^−1^ with 32 co-added scans, in the wavelength range from 16,667 to 2500 nm. Background spectra of the clean ATR sampling surface were collected each time before the collection of samples’ spectra using the same acquisition parameters. The MIR spectra acquisition is similar to a previous work [37].

### 2.4. Spectral Pre-Processing and Chemometrics

Pre-processing of a spectrum is an essential step to eliminate non-chemical biases such as background noise and light scattering effect. In this study, SIMCA^®^ was used for the pre-processing and data analysis. SIMCA^®^ from Umetrics is a software that offers principal component analysis (PCA), an unsupervised multivariate analysis, and orthogonal partial least square-discriminant analysis (OPLS-DA), a supervised multivariate analysis.

PCA is an unsupervised technique that reduces the dimensionality of datasets while retaining the most important information, allowing trends within the multivariate datasets to be observed. This technique gives us a rough idea about the variation between the samples without prior knowledge of sample grouping, but PCA is not able to classify samples according to known classes. The application of PCA can be found in previous studies, especially in food adulteration [29,30,38,39]. In the SIMCA^®^ algorithm, PCA is performed on all classes of data to obtain the principal component that is responsible for the most variation in the class. Then, some part of the data is taken out to perform cross-validation with a different number of principal components on the rest of the data until every datum has been taken out once. As an example, when the default setting of SIMCA^®^ is assigned the cross-validation groups as 7, the 7th of the observations is excluded during each cross-validation round, basically highlighting that the number of cross-validation groups decides the size of the excluded observation group during component computation. In the end, a prediction model with the lowest prediction error and an optimum number of principal components is produced. In order to obtain an overview of the dataset, a few (2 or 3) principal components are often sufficient. However, if the PC model is used for modelling or for other predictions (e.g., principal properties), cross-validation (CV) should be used for testing the significance of the principal components. In this way, the “significant” number of PC components, A, is obtained, which is essential in modelling.

OPLS-DA is a further analysis to the PCA, as this method is capable of resolving the cluster of data in the direction of maximum class separation which enables the discrimination between different classes of samples. The model generated can be interpreted with specific indicators such as goodness of fit of data and the predictive ability of the model. The predictive ability of a model is based on predictive score, whereby the predictive score, t, is obtained for each subject, and the value is then used to plot the score plot. Conversely, the 7-fold cross-validation is carried out to determine the stability of the sample point in the model. The cross-validation test is an alternative to the training and test set in OPLS-DA. First, the samples are split into 7 different subsets by default. Each time, 6 subsets were trained and tested with the remaining 1 subset. This process is repeated by default until the predictiveness of the model, Q^2^, is above the acceptance significance limit (0.01). This method is useful to discriminate samples of different classes such as authentic and adulterated samples. The application of OPLS-DA can be found in food adulteration-related studies [40,41,42]. The combination of PCA and OPLS-DA are complementary to each other. The different sources (plant based or animal based) of adulterants such as karaya gum and pork skin used for EBN adulteration [5] indicate that they serve different purposes. Due to their different characteristics, each type of adulterant is consistently applied in any batch of EBNs. Therefore, in this study, the discriminant models were performed separately according to adulterant type to mimic the real-world scenario.

#### 2.4.1. Spectral Pre-Processing on Near-Infrared Spectrum

The whole range VIS/SW-NIR spectra (740–1070 nm) were pre-processed with Standard Normal Variate (SNV) using SIMCA^®^. The SNV operation subtracts each value in the specific column by the mean value and is divided by the standard deviation value of the corresponding column. This operation aims to reduce spectral noise and background effects.

#### 2.4.2. Multivariate Analysis on Near-Infrared Spectrum

The qualitative analysis for the authentication of EBN samples was carried out using chemometrics. In SIMCA^®^, the authentication analysis was performed by using the OPLS-DA algorithm. The algorithm analysed all the samples involved and subsequently performed the cross-validation and formed the best model of EBN authentication.

#### 2.4.3. Spectral Pre-Processing on Mid-Infrared Spectrum

The pre-processing of MIR spectra (16,667–2500 nm) with SIMCA^®^ was similar to the pre-processing employed on the VIS/SW-NIR spectra.

#### 2.4.4. Multivariate Analysis on Mid-Infrared Spectrum

The procedure of SIMCA^®^ to detect EBN authenticity using VIS/SW-NIR data was employed similarly to the extracted MIR data.

## 3. Results and Discussions

### 3.1. Spectral Analysis

Figure 3A shows the overlay of VIS/SW-NIR spectra (740–1070 nm) absorbance of authentic EBN and pure adulterant. The spectra shown in Figure 3B is the inset of an individual scan of one authentic EBN and adulterants. The adulterant samples include the common structural adulterants, karaya gum, gelatine, nutrient agar, melamine and collagen. Gelatine VIS/SW-NIR spectrum is distinctly different from the rest of the adulterants with the highest intensity, whilst the spectra of other adulterants are similar to authentic EBNs with a much lower absorbance intensity. There is variation between authentic EBN as seen in the difference in intensity of VIS/SW-NIR spectra.

The absorption bands of these SCiO^TM^ VIS/SW-NIR spectra are the overtones of functional groups with their associated chemical groups. The band assignment of VIS/SW-NIR is summarised in Table 2. The absorption bands around 840 nm are the third overtone of the hydroxyl (O-H) [43,44] and fourth overtone of the carbonyl group (C=O). The amino group (NH_2_), which indicates the presence of protein, can be observed at the absorption bands around 775 to 825 nm and 1020 to 1050 nm [45]. Seeing the tangible differences between the spectrum of EBN plotted against the spectrum of the different adulterants, further investigation was carried out to explore these differences by adulterating the authentic EBN with the different adulterants at low levels and comparing their respective spectra, thus at the same time, approximating the technique’s limit of detection.

From the visual comparison of the VIS/SW-NIR spectra, the spectrum of authentic EBN differs from those of adulterated EBNs, in terms of intensity, as shown in Figure 4A–E). The VIS/SW-NIR spectra consist of individual scans of each authentic EBN and adulterants. In general, the absorbance intensity of adulterated EBNs is lower than authentic EBNs and is closer to the authentic EBNs, compared to the pure adulterants. Class D (collagen) shows the most differences, whilst class C (gelatine) shows the least differences in intensity between authentic EBN and adulterated EBNs. The bands representing adulterated EBNs found across the spectra are higher or lower in intensity, varying with the adulteration percentage, as indicated in Figure 4. Some of the spectra of adulterated EBNs do not show a proportional increase in intensity as the adulteration percentage increases. This is probably caused by the inhomogeneity in the samples. However, the shape and peaks found on the spectra are almost identical. The spectra do not reveal peaks that are distinctively different between authentic and adulterated samples. As expected, the spectra of pure adulterants are more distinguishable from the authentic and adulterated EBNs. This could indicate that the functional groups of compounds in adulterants are different from that of authentic EBN. By comparison, the adulterated samples have a lower concentration of adulterants (1, 5 and 10%), which makes the spectra of adulterated EBNs more similar to authentic EBN. Hence, it may be difficult to detect the presence of adulterants in the EBN by visual inspection of the spectra, which makes chemometric analysis useful and crucial in detecting adulteration at low levels for VIS/SW-NIR work. Taking note of this observation, PCA and multi-class discriminant analysis were employed for further investigation.

Similar to the VIS/SW-NIR work, the mid-infrared reflectance spectrum of authentic EBN was compared with the spectrum of each type of adulterant, and these are shown in Figure 5, covering the absorption bands ranging from 16,667 to 2500 nm. The bands represent certain functional groups that are present in EBN, and the information is summarised in Table 3. The bands around 7576, 7143 and 6944 nm show the presence of the amine group (C-N), aldehyde group (CH=O) and carboxylic group (COOH), respectively, [46] while the presence of protein is indicated by the absorption bands around 6494 and 6098 nm.

In the same figure, an intense peak which represents the polysaccharide at 9551 nm in EBN can be observed around 9699 nm in adulterants. Two peptide-related peaks found around 6120 nm (contributed by N-H) and 6540 nm (contributed by C=O) [47] in EBN are present in melamine and collagen, but for karaya gum and gelatine, only the peak at 6120 nm was found with some shift in peak position. Conversely, there is no such peak found in the spectrum of nutrient agar which indicates that peptides are absent in nutrient agar. The protein (peptide) and carbohydrate (polysaccharide) related peaks show the highest intensity in EBN, as these compounds have the highest weightage in EBN [48]. The vibrational band at 3064 nm due to the presence of O-H bonds (water) is found in EBN, with higher intensity compared to the same peak found in adulterants, and this may suggest that EBN moisture content is relatively higher.

Additionally, the MIR spectra also show that the melamine results differ the most from that of authentic EBN as well as the rest of the adulterants. At wavelength 6998 nm, an intense peak that represents the aromatic amine functional group, C-N(H_2_), is present in melamine. Six other peaks are associated with N-H_2_ (amino group) [49] present at 2883, 2925, 3007, 3120, 8496 and 12345 nm in the melamine spectrum. These peaks which are associated with the amino-functional group are not found in EBN and other adulterants. MIR spectra thus enable melamine to be differentiated from authentic EBN and other adulterants. Conversely, there are a few peaks that are present and found only in the karaya gum spectrum. At 2814 nm, the peak found is related to the C=O group of carboxylic acid [50], whilst at 8051 nm, the peak present is related to the PO_2_^−^ group in nucleic acid [51]. Peaks that are associated with C-H deformation (holocellulose) at 7299 nm and C-H_2_ scissoring vibration of cellulose at 7082 nm are also identified [52]. Another small peak at 8734 nm related to the C-O-C functional group from polysaccharide is also found in karaya gum [53]. All the peaks unique to karaya gum are probably due to the origin of the material, which is extracted from trees and contains namely galactose, rhamnose and gluconic acids. In the spectrum of gelatine, there are two major peaks identified near 6120 nm (N-H) and 9551 nm (polysaccharide). In the region of 7143 nm to 8333 nm, a few minor convoluted peaks are present. These peaks belong to the C-H_2_ vibrational band (cellulose) and C-N functional group of amide III, [54]. Conversely, a small peak at 7163 nm which represents the C-O group of carboxylic acid [55] and 9699 nm which is associated with polysaccharide are found in nutrient agar spectrum. However, the rest of the nutrient agar spectrum is relatively flat, which can be explained by the lack of protein in the material; thus, its spectrum can be differentiated from that of EBN. Lastly, the spectrum of collagen has the highest similarities with EBN, albeit the intensity of the peaks is much lower across the board as compared to the peaks in the EBN spectrum. It is also observed that a small peak at 9259 nm which is associated with PO_2_^−^ functional group (phospholipid) [51] is found in collagen but not in EBN. Thus, following these findings, further work was performed to explore the differences between authentic EBN against EBN adulterated with the different adulterants at low levels to test the technique’s limit of detection.

Figure 6 shows the MIR spectra of authentic EBN plotted against adulterated EBNs at different levels of adulteration, which are 1, 5 and 10%. The MIR spectra of adulterated EBNs are almost similar, except for adulterated EBNs in class A (melamine), where there are small peaks present around 2900 nm. The similarities in MIR spectra of authentic EBN plotted against the spectra of EBN samples adulterated with the different adulterants are consistent with the findings in the VIS/SW-NIR analysis in the present work. Through visual inspection, it can be highlighted that those pure adulterant spectra when plotted against authentic EBN spectrum are much more distinct in terms of peak shape and intensity, as compared to adulterated EBNs spectra plotted against the authentic EBN spectrum.

The similarities and differences between EBN and the adulterants in terms of chemical composition indicate that the ability to detect adulteration in EBN may differ for both VIS/SW-NIR and MIR techniques. Gelatine (VIS/SW-NIR) and melamine (MIR) show distinct differences in IR bands when compared to authentic EBN, hence the differentiation between them is expected to be easily performed. Overall, the visual inspection of both VIS/SW-NIR and MIR spectra indicates that there are marked differences between authentic EBN and the pure adulterants, but they are less obvious between authentic and adulterated EBNs. Hence, it is imperative to have the findings from both techniques further explored using chemometrics, starting with PCA and followed by multi-class discriminant analysis.

### 3.2. Adulteration Classification

#### 3.2.1. PCA and OPLS-DA of Near-Infrared Spectrum

In Figure 7, the principal component analysis was performed to observe the variation, trend or overlapping across the samples without class separation. The spectra pre-processing applied was the standard normal variate (SNV). There was a clear separation between the pure adulterants and the pure or adulterated EBNs. The samples outside the 95% confident interval circle were significantly different from those inside the circle. All of the adulterants plotted outside of the circle indicate that adulterants are different from EBNs in terms of the chemical composition. However, there are some authentic EBNs which fall outside of the circle. This observation could be due to the differences caused by geographical origin [36].

Figure 8 illustrates a three-class OPLS-DA model constructed on the same samples. This shows a clearer separation between the classes of adulterants, pure EBNs and adulterated EBNs, compared to the PCA plot. The pure adulterants were once again reasonably clearly differentiated from the genuine and adulterated EBNs. For four of the five adulterants, the adulterated EBN samples could be differentiated from the genuine EBN samples. However, for the karaya gum adulterant, the discrimination between the pure EBNs, pure adulterants and the adulterated EBNs was not evident, indicating that the model has limited discrimination power for karaya gum compared to the other types of adulterants. This suggests more similarities in chemical composition between Karaya gum and the EBN. The 1 and 5% adulteration samples were mingled with the pure EBN samples, and the 10% adulteration sample was close to the cluster of pure EBNs. In the case of the melamine adulterant class, the lowest discrimination limit is 5%. From the plot, the gelatine adulterant class also exhibits a low discrimination rate, as the adulterated samples were almost indistinguishable from the pure EBNs class. Therefore, as might be expected, the lower the adulteration percentage, the weaker the discrimination power of the model.

Since the purpose of this study was to differentiate the authenticity of EBNs and develop a rapid screening test for the point of use testing, the model was simplified to a two-class OPLS-DA model using VIS/SW-NIR data, as shown in Figure 9. The pure adulterants and adulterated EBNs were combined into one class and plotted against the authentic EBNs. In the figure, the cross-validation (CV) score plot (left) is displayed side by side with its complementary scores scatter plot (right) to visualize the stability of a single point as well as a group. An observation showing the two classes placed in the opposite direction and separated from each other is ideal. In the present study, gelatine, collagen and nutrient agar exemplified this, but for melamine and karaya gum, the pure EBN samples overlap with the adulterated class; hence, the results indicated that the class assignment was uncertain.

To further validate the observations of the two-class model, the loading S-plot in Figure 10 was used to visualise both the covariance and the correlation structure between the X-variables and the predictive score t [1]. The X-variables situated far out on the wings of the S-plot combined high model influence with high reliability and are of relevance in the search for markers. Each matrix showed different markers as in the case of non-authentic EBN samples. The numbers in the plot denote the observed wavelengths of the VIS/SW-NIR spectra of the EBNs. The summary of the description of the denoted wavelengths was presented earlier in Table 2. The information about the functional groups represented by the VIS/SW-NIR wavelengths was limited. We can conclude that the discriminating factor for class A (melamine), class B (karaya gum) and class E (nutrient agar) was due to the third harmonic of the alkyl groups. However, in class C (gelatine) and class D (collagen), the third harmonic of the amine group and the second harmonic of the hydroxyl group were responsible for the discriminant power of the model.

Another option of identifying the predictive ability of the model is by developing a misclassification table (CV method), which is presented in Table 4. In the table, the results are presented in the form of a percentage of correct classifications for a particular class. The total number of samples, the total number of correct and incorrect classifications, and the classification of 1, 5 and 10% adulterated EBNs are also displayed. The collagen and nutrient agar class had 100% correct classification for both authentic EBNs and non-authentic EBNs. Meanwhile, 100% correct classification for authentic EBNs was achieved for the melamine, karaya gum and gelatine classes. The karaya gum class had the lowest correct classification rate of non-authentic EBNs at 40%. It was relatively difficult for the model to determine the authenticity of the samples with low adulteration percentage, 1% and 5%. However, the low *p* value (<0.05) indicated that the models were statistically significant classification models.

#### 3.2.2. PCA and OPLS-DA of Mid-Infrared Spectrum

The same procedures to obtain classification models using the VIS/SW-NIR data were also performed on the MIR spectra. In MIR spectra, the pre-processing was undertaken using mean-centring. The results of the PCA are shown in Figure 11. Generally, the pure EBN class and adulterated EBNs showed similar trends, in which most samples were retained inside the circle with some overlap, a similar pattern that was observed with the VIS/SW-NIR data. Conversely, there was a distinct separation of the pure adulterant cluster from that of pure and adulterated EBNs.

Additionally, three-class OPLS-DA was performed on the same samples, and the results are presented in Figure 12. In this supervised method, the groupings of the three classes were obvious for the melamine and karaya gum adulterants. However, the other three types of adulterants were not clearly separated, and some of the adulterated EBN samples were grouped with pure EBNs. This indicates that adulterated EBNs possessed a certain detection limit for the samples to be discriminated. It can be observed that in the nutrient agar adulterant class, the lowest detection was 5% adulteration. Nevertheless, in the gelatine adulterant class, only 10% adulteration can be identified. The model for the collagen adulterant class possessed the weakest discriminant power, as none of the adulterated samples can be separated from the pure EBNs, implying a similar chemical structure between collagen and EBN. Similar findings have been reported previously by Adenan [36].

The same two-class approach utilized for the VIS/SW-NIR was applied to the MIR data. The pure adulterants and adulterated EBNs were combined into one class (non-authentic EBNs) and plotted against the authentic EBNs. In the two-class classification in Figure 13, similar to the SCiO^TM^ VIS/SW-NIR analysis, the cross-validation score plot (left) was displayed together side by side with the scores scatter plot (right) to assess the stability of a single point as well as classes. Overall, some of the adulterated samples were grouped with the pure EBNs (gelatine, collagen and nutrient agar), and these observations indicate that the class assignment was not certain. Nevertheless, a similar distribution for the samples suggests that the models are stable, and the inclusion or exclusion of a certain observation is less likely to affect the model.

The loading S-plot of the two-class OPLS-DA using MIR spectra was presented in Figure 14. Each matrix showed different markers as in the case of non-authentic EBN samples. The summary of the description of the denoted wavelength is presented in Table 3. Each of the matrices has different influencing factors that contribute to the discrimination of authenticity. In class A, representing the melamine adulterant, the carboxyl group of the amino acid and moisture content were responsible for the grouping of samples. In class B, karaya gum, the presence of a tri-substituted benzene ring and an alkyl group in aldehyde contributed to the separation of authentic and adulterated samples. Amine and mono-disubstituted alkyl groups were found to be important factors in class C, representing the gelatine adulterant. These functional groups did not differ much from the EBN’s functional groups; hence, discrimination may prove to be difficult. For class D (collagen), the discriminating factors were the tri-substituted benzene ring, and the carbon double bond in the carboxyl and alkyl groups in carbohydrates. For nutrient agar adulterant, class E, the presence of carboxyl and alkyl groups seemed to be the factors influencing the discrimination of authentic EBNs from that of the adulterated ones.

Similarly, as was carried out on the VIS/SW-NIR data, a misclassification table was also developed for the MIR data to investigate the predictive ability of the model. In Table 5, the results are presented in the form of percentage and the proportion of correct classification for a particular class. The total number of correct classifications is also displayed. All of the classes had a 100% correct classification for authentic EBNs. Conversely, only the karaya gum class had 100% correct classification for adulterated EBNs. The nutrient agar had the highest correct classification of the adulterated EBNs, which was 40%, followed by class C (gelatine) with 75% correct classification and class D (collagen) with 40%. In the collagen class, 1 and 5% of adulterated samples could not be recognised, while in the gelatine class, 1% of adulterated samples was misclassified as authentic EBNs. The presence of the amino group (N-H) and C-H band from carbohydrates had great impact on the discriminant model, such that class C (gelatine) and D (collagen) showed higher similarities in the MIR spectra, which weakened the capability of the model in differentiating the authenticity of EBN. Although melamine was thought to have given rise to the apparent protein content, the amino group in melamine differed significantly from that of EBN. Lastly, the low *p* value (<0.05) indicated that they are a significant classification model.

### 3.3. Comparison of Handheld VIS/SW-NIR against a Benchtop MIR

From Table 6, the overall performance of VIS/SW-NIR (sensitivity and specificity) is better than that of MIR. The performance of the specific adulterants varies between these two techniques. For instance, the sensitivity of the Karaya gum adulterant class is the lowest in VIS/SW-NIR, 40%, but is the highest in MIR, 100%. Conversely, the sensitivity of the collagen adulterant class is the lowest in MIR, 40%, but the highest in VIS/SW-NIR, 100%. Both devices provide good results, and it shows that the different range of the infrared spectrum has different contributions. Previously, the study performed by Adenan [36] had been performed differently from the methodology of this study; hence, only the overall specificity is known, which is 61%. In that study, the findings are similar to the current work, as adulteration at 1% is hard to detect. The results of the previous study cannot be compared directly with the current work. Adulteration at 1 and 5% is relatively harder to detect by these techniques. The VIS/SW-NIR technique has the advantage of mobility, smaller power supply, shorter scanning acquisition time and requiring less environmental control compared to the MIR technique. Hence, a deeper study on the usage of VIS/SW-NIR in EBN adulteration is needed, as this hand-held VIS/SW-NIR has great potential.

The sample size of authentic, adulterated samples and adulterants, a total of 77 samples, may be improved by obtaining more authentic samples. This is generally not possible because of their high cost and rarity. Moreover, the hand-held VIS/SW-NIR sensor is the first generation of the pocket-size spectrometer and has a limited range of wavelength/absorption band, and improvement is expected in the future. Nevertheless, the results from this study show promise as a new approach for the rapid screening of EBN authenticity, with a low-cost handheld portable sensor for point of use decision making for this important and expensive product.

## 4. Conclusions

In summary, infrared spectroscopy coupled with multi-class discrimination analysis has the potential to be implemented in the authentication work of EBNs. The results of authentication tests using VIS/SW-NIR showed that collagen and nutrient agar adulterated EBNs had no misclassification. As for the rest of the classes, the successful classification rate was followed by the melamine class at 99.1%, gelatine class at 98.18% and karaya gum class at 94.4%. Conversely, the MIR authentication model had no misclassification on karaya gum adulteration. The second highest correct classification rate was the melamine class at 99.1% followed by the nutrient agar class, 98.2%; gelatine class, 97.3%; and collagen class at 94.4%. Conclusively, the VIS/SW-NIR technique applied using a handheld device proved to be promising in terms of EBN adulteration screening performance against the more reliable benchtop MIR, thus providing a bright and promising outlook in food adulteration screening from a field-deployable context. However, this VIS/SW-NIR technique should only be used as a screening tool, and a proper confirmation test should be performed in a laboratory for any adulterated or random samples. In the future, a larger batch of EBN samples taking into consideration geographical and weather variance should be included. The same goes for the variation of adulterants involved since adulteration activities may change from time to time.

## Figures and Tables

**Figure 1 foods-11-02401-f001:**
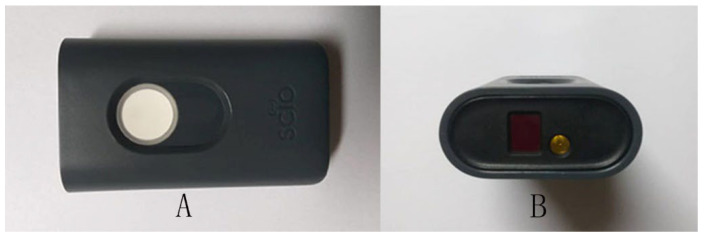
Consumer Physics VIS/SW-NIR molecular sensor, SCiO^TM^ (**A**) VIS/SW-NIR spectrometer, (**B**) The location of light source and detector).

**Figure 2 foods-11-02401-f002:**
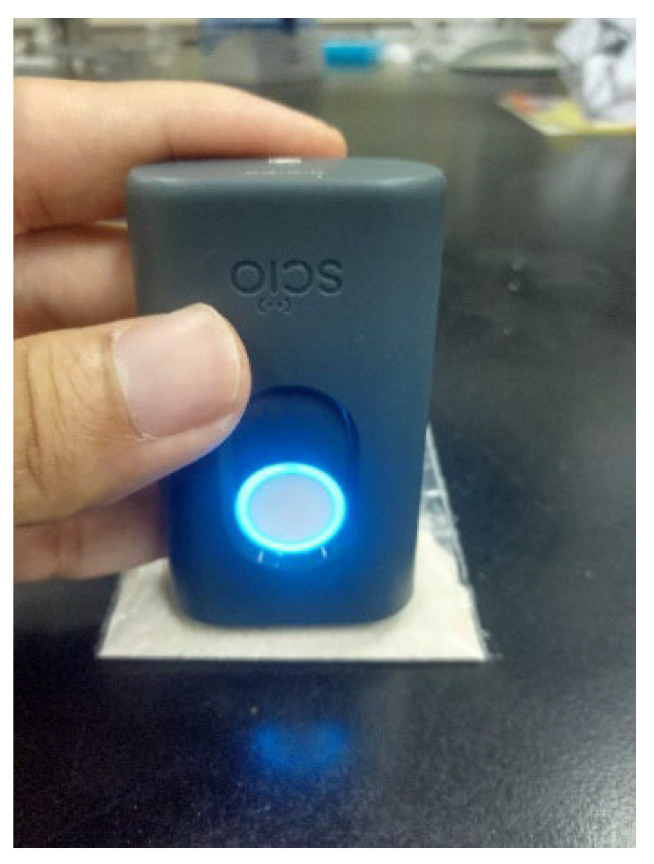
The SCiO^TM^ VIS/SW-NIR sensor spectral data acquisition.

**Figure 3 foods-11-02401-f003:**
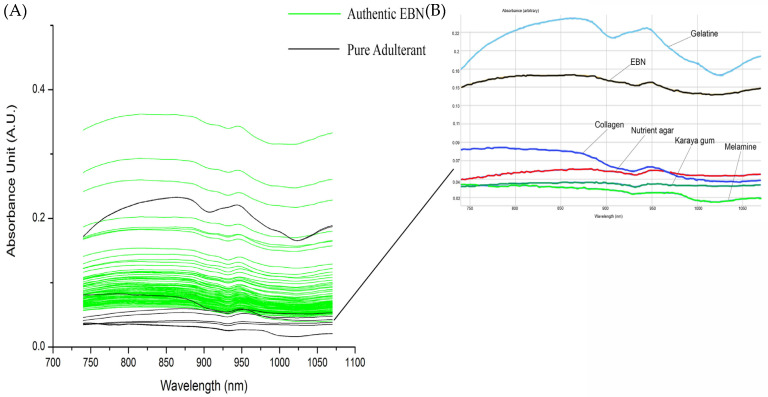
The overlay of VIS/SW-NIR spectra of authentic EBN with pure adulterants (the inset of overlay of VIS/SW-NIR spectra of authentic EBN and adulterants).

**Figure 4 foods-11-02401-f004:**
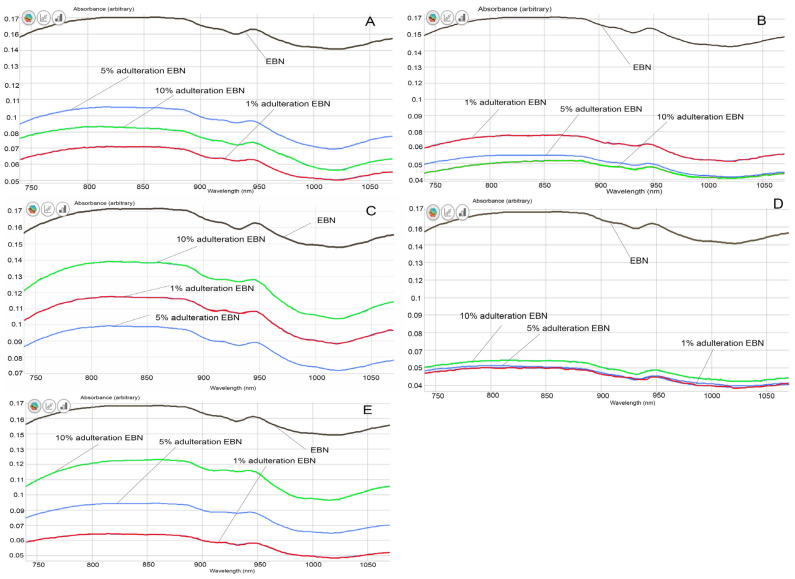
The VIS/SW-NIR spectra of authentic EBNs and adulterated EBNs at concentrations of 1%, 5% and 10% ((**A**) melamine, (**B**) karaya gum, (**C**) gelatine, (**D**) collagen, (**E**) nutrient agar).

**Figure 5 foods-11-02401-f005:**
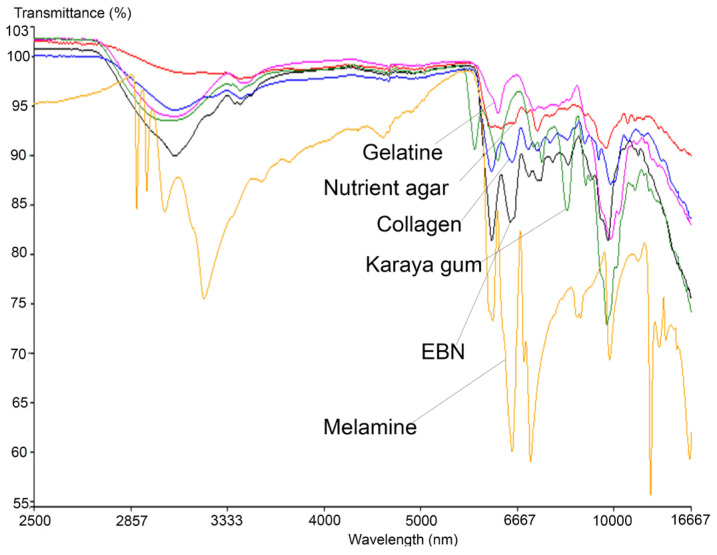
The overlay of MIR spectra of authentic EBNs with pure adulterants.

**Figure 6 foods-11-02401-f006:**
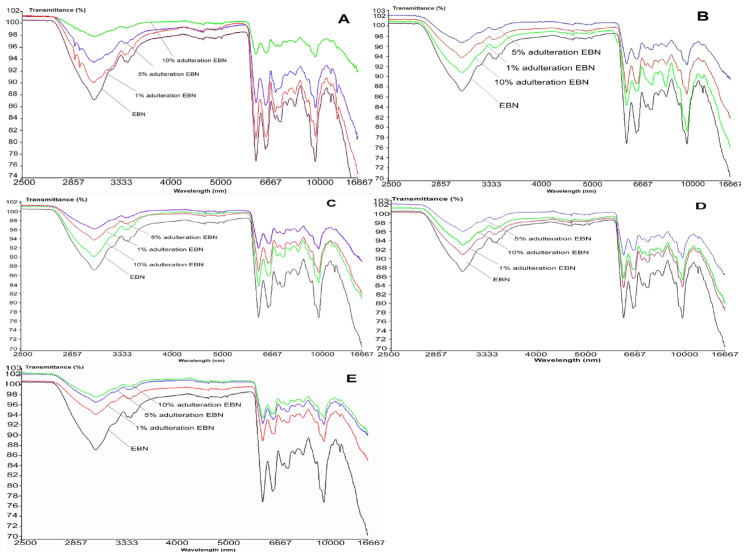
The MIR spectra of authentic EBN and adulterated EBNs at concentrations of 1, 5 and 10% ((**A**) melamine, (**B**) karaya gum, (**C**) gelatine, (**D**) collagen, (**E**) nutrient agar).

**Figure 7 foods-11-02401-f007:**
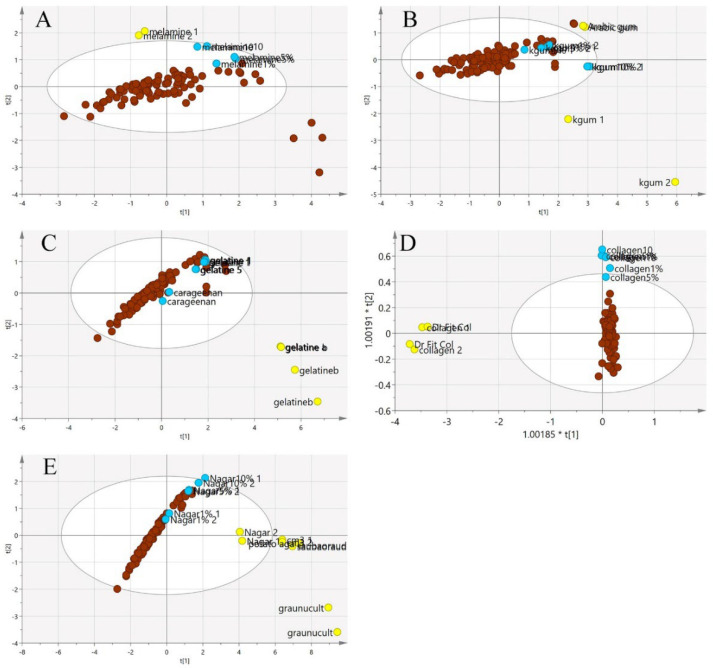
The three-class PCA-X score model is based on VIS/SW-NIR spectra ((**A**) melamine, (**B**) karaya gum, (**C**) gelatine, (**D**) collagen and (**E**) nutrient agar). The authentic class is pure EBN samples, the adulterant class is the material for adulteration, and the adulterated sample is the EBNs with 1, 5 and 10% of an adulterant. The distribution of the class shows that the adulterant class had the most variation from the others (authentic: 

, adulterant: 

, adulterated: 

).

**Figure 8 foods-11-02401-f008:**
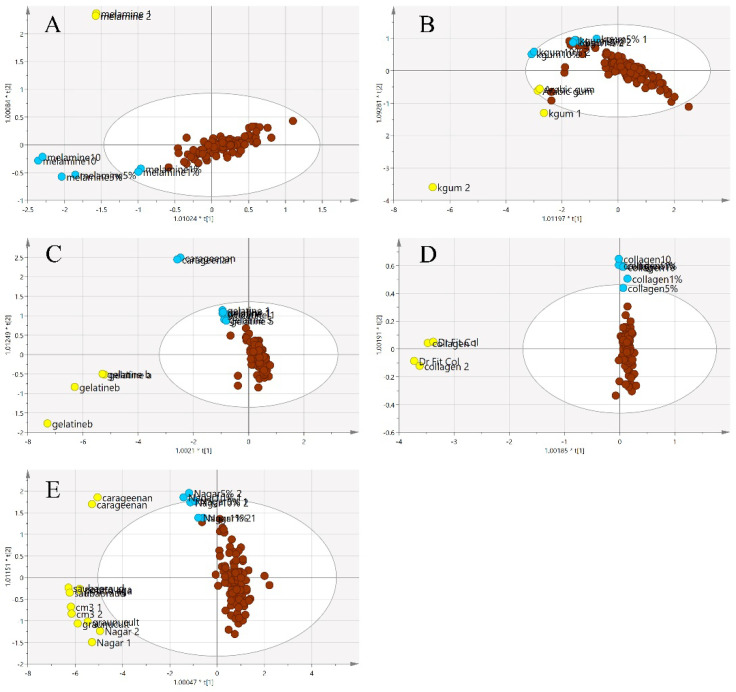
The three-class OPLS-DA score model for authentic EBNs and adulterant and adulterated samples based on VIS/SW-NIR spectra ((**A**) melamine, (**B**) karaya gum, (**C**) gelatine, (**D**) collagen and (**E**) nutrient agar). This model classified the samples into three different classes, and the distinction of classes was obvious between adulterants and the others. The adulterated samples exhibit more similarities with authentic EBNs, as there was some overlap in all of the models except for model E, where the separation of three classes was clear (authentic: 

, adulterant: 

, adulterated: 

).

**Figure 9 foods-11-02401-f009:**
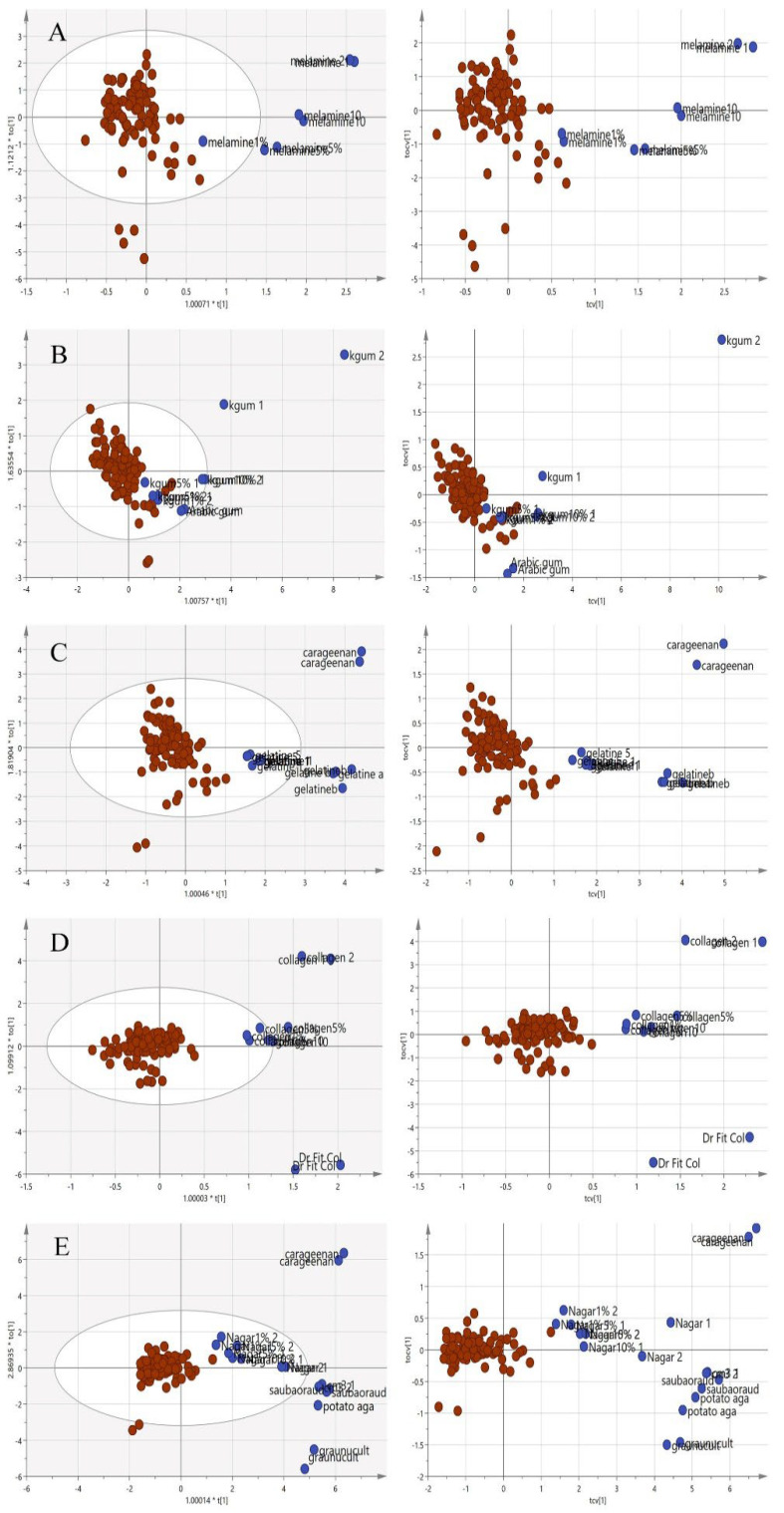
Each tier of (**A**–**E**) consists of a two-class OPLS-DA score model plotted side by side with the CV score model using VIS/SW-NIR spectra ((**A**) melamine, (**B**) karaya gum, (**C**) gelatine, (**D**) collagen and (**E**) nutrient agar). The overlap of some points from both classes in the CV score model indicated that the assignment of the class was uncertain (authentic: 
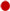
, non-authentic: 

).

**Figure 10 foods-11-02401-f010:**
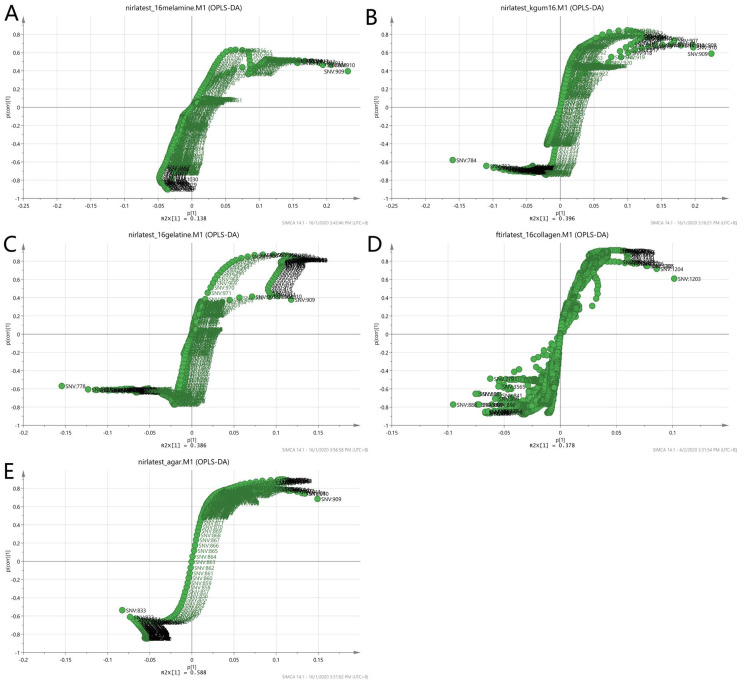
Loading S-plot of a two-class OPLS-DA model for authentic and non-authentic classes using VIS/SW-NIR spectra ((**A**) melamine, (**B**) karaya gum, (**C**) gelatine, (**D**) collagen and (**E**) nutrient agar) The number represents the wavelength of the VIS/SW-NIR spectra, which is the significant factor that contributed to the discrimination of the two classes.

**Figure 11 foods-11-02401-f011:**
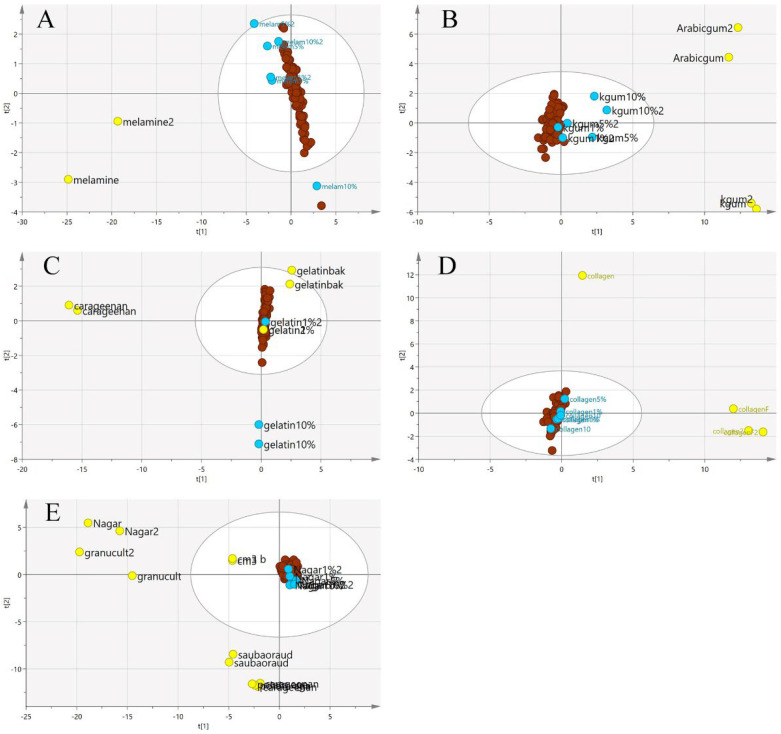
The three-class PCA-X score model of EBN samples adulterated with five different adulterants ((**A**) melamine, (**B**) karaya gum, (**C**) gelatine, (**D**) collagen and (**E**) nutrient agar) based on MIR spectra. The plots show that the adulterated samples and authentic EBNs had less variation than pure adulterants (authentic: 

, adulterant: 

, adulterated samples: 

).

**Figure 12 foods-11-02401-f012:**
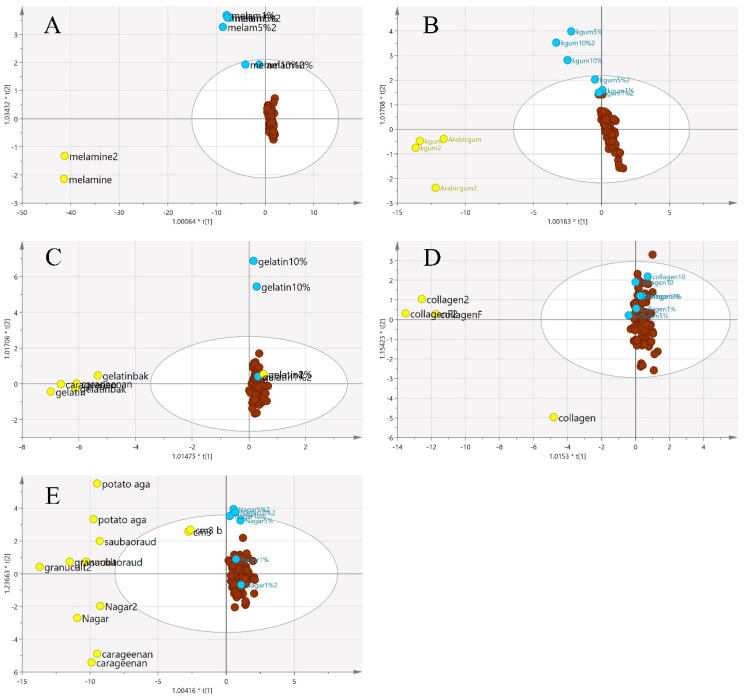
The three-class OPLS-DA score model with MIR spectra ((**A**) melamine, (**B**) karaya gum, (**C**) gelatine, (**D**) collagen and (**E**) nutrient agar). A clearer distinction is evident between the three classes except for model D. This is a supervised classification method where the samples were grouped with the most similarities (authentic: 

, adulterant: 

, adulterated: 

).

**Figure 13 foods-11-02401-f013:**
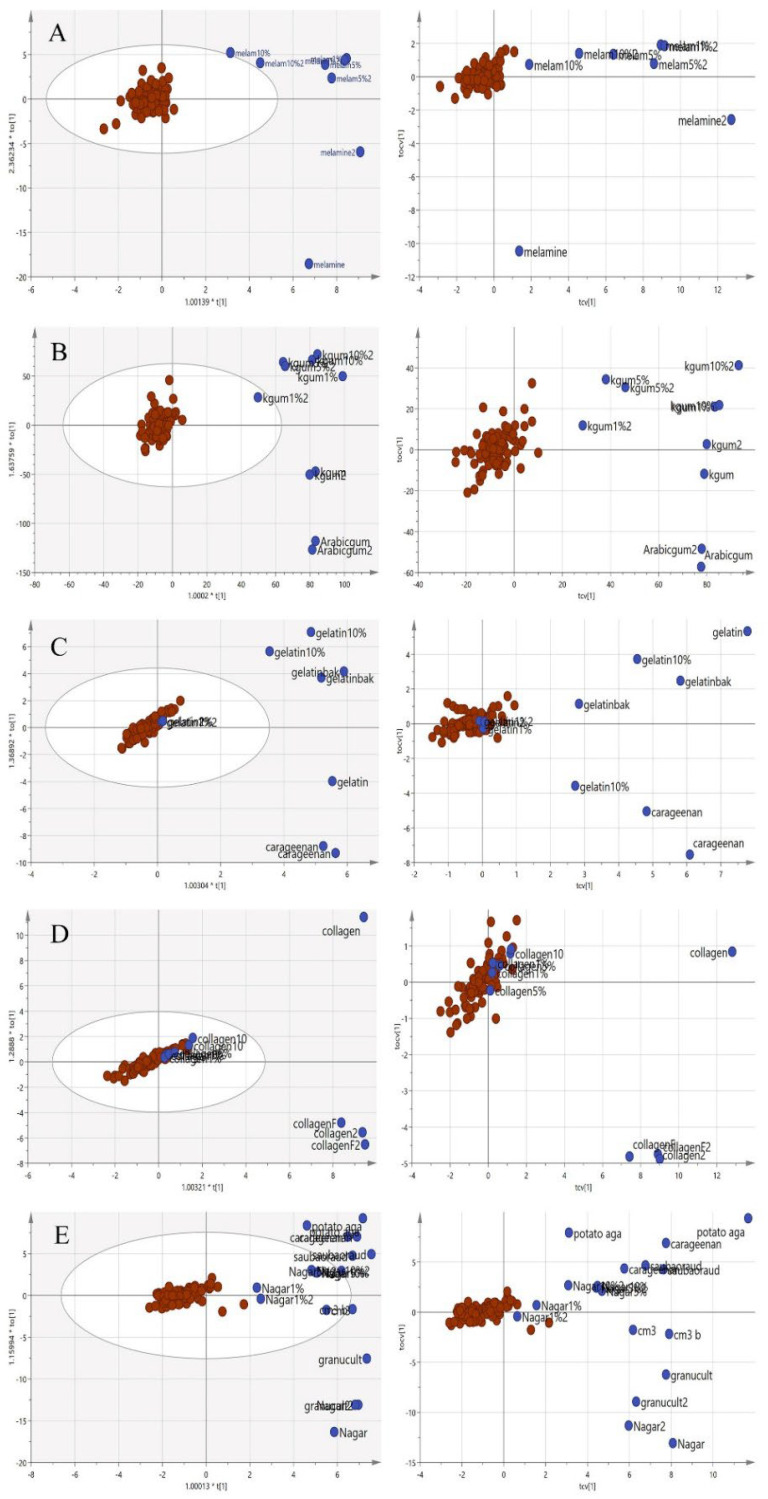
The OPLS-DA and CV score plot of two-class classification using MIR spectra ((**A**) melamine, (**B**) karaya gum, (**C**) gelatine, (**D**) collagen and (**E**) nutrient agar). The score model demonstrated that the model was stable and significant, as there was a clear separation of samples, although there was a mild overlap between the two classes (authentic: 
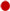
, non-authentic: 

).

**Figure 14 foods-11-02401-f014:**
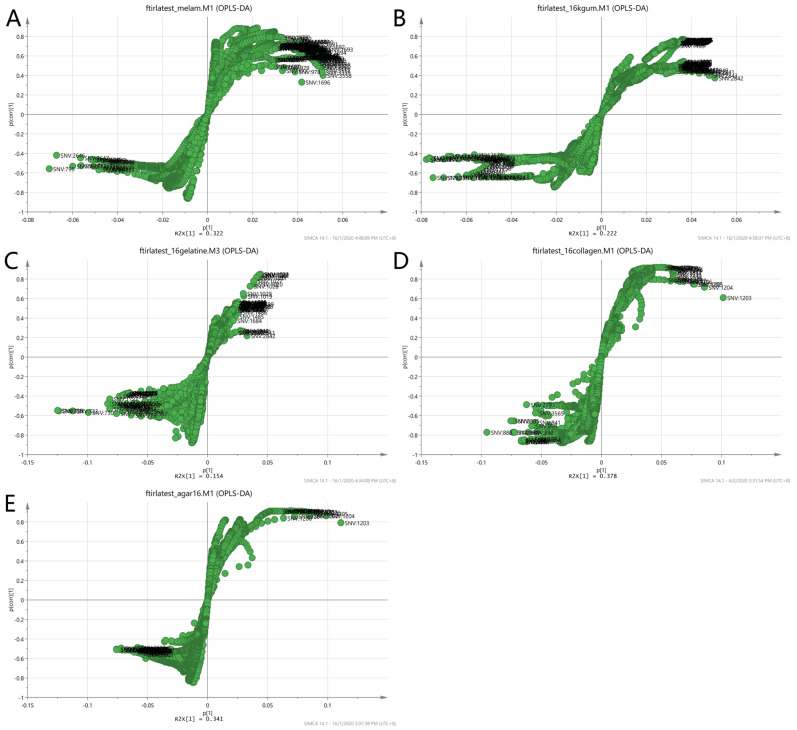
The S-plot of the two-class OPLS-DA model of the variables among the samples using MIR spectra ((**A**) melamine, (**B**) karaya gum, (**C**) gelatine, (**D**) collagen and (**E**) nutrient agar). Each point represents the wavelength of the spectra. The plot at the outermost is the factor contributing to the grouping of the samples.

**Table 1 foods-11-02401-t001:** The origins and date of collection of EBNs.

States	Locations	Year
Terengganu	Kuala Terengganu	2015/2016
Sarawak	Betong, Kuching, Miri, Mukah, Samarahan, Sibu	2015/2016
Sabah	Beauford, Kudat, Tongod, Tuaran	2015/2016
Pahang	Bandar Tun Razak, Kuantan	2015/2016
Negeri Sembilan	Kuala Pilah, Port Dickson, Seremban	2015/2016
Kelantan	Gua Musang	2015/2016
Johor	Batu Pahat, Kuala Rompin, Kulai, Masai, Muar, Pontian Kecil, Senai	2015/2016

**Table 2 foods-11-02401-t002:** The functional groups in EBN are denoted by the wavelength of VIS/SW-NIR bands and their respective harmonic level.

Wavelength (nm)	Functional Group	Harmonic
770–790	R-NH_2_, R-NH-R	3rd harmonic
835–840	O-H	3rd harmonic
850–860	Aromatic	3rd harmonic
900–920	CH_3_	3rd harmonic
910–950	O-H	2nd harmonic

**Table 3 foods-11-02401-t003:** The functional groups in EBN are denoted by the wavelength of MIR bands and the associated possible compounds.

Wavelength (nm)	Functional Group	Compound
13,889–13,514	C-H	mono-di-substituted
13,514–13,158	C-H	tri-substituted benzene
13,158–12,579	C-H	alkane
11,236–11,111	C-H	tri-substituted benzene
10,309–10,000	HC=CH	alkene
9901–9709	C-N	amine
8333–8197	CH_2_, C-O	vinyl ether
6757–6711	N-O	nitro compound
6757–6667	CH_2_	methyl group, alkane
5917–5882	C=O	ester acid
3788–3774	S-H	thiol
3534–3509	C-H	aldehyde
2817–2809	O-H	alcohol

**Table 4 foods-11-02401-t004:** Misclassification tables from a two-class OPLS-DA model applied to a prediction set of authentic and non-authentic EBNs using VIS/SW-NIR spectra (A: melamine, B: karaya gum, C: gelatine, D: collagen and E: nutrient agar). In the first row, the first column is the classes of the model, and the last column is the *p* value (<0.05) which represents the significance of the classification model. The second and third column is the total number of samples in each class and their respective individual number of correct classifications. The last three rows indicate the correct classification of adulterated EBNs into non-authentic EBNs or into incorrect classification of authentic EBNs class.

**A**	**Authentic**	**Non-Authentic**	**Total**	***p* Value**
Quantity	98	8	106	2.85 × 10^−19^
Correct classification	100%	75%	99.1%	
1% adulteration	/			
5% adulteration		/		
10% adulteration		/		
**B**	**Authentic**	**Non-authentic**	**Total**	***p* value**
Quantity	98	10	108	4.51 × 10^−11^
Correct classification	100%	40%	94.4%	
1% adulteration	/			
5% adulteration	/			
10% adulteration		/		
**C**	**Authentic**	**Non-authentic**	**Total**	***p* value**
Quantity	98	12	110	1.81 × 10^−29^
Correct classification	100%	83.3%	98.18%	
1% adulteration		/		
5% adulteration	/			
10% adulteration		/		
**D**	**Authentic**	**Non-authentic**	**Total**	***p* value**
Quantity	98	10	108	1.12 × 10^−15^
Correct classification	100%	100%	100%	
1% adulteration		/		
5% adulteration		/		
10% adulteration		/		
**E**	**Authentic**	**Non-authentic**	**Total**	***p* value**
Quantity	98	16	114	5.16 × 10^−35^
Correct classification	100%	100%	100%	
1% adulteration		/		
5% adulteration		/		
10% adulteration		/		

**Table 5 foods-11-02401-t005:** Misclassification table from a two-class OPLS-DA model applied to a prediction set of authentic and non-authentic EBNs using MIR spectra (A: melamine, B: karaya gum, C: gelatine, D: collagen and E: nutrient agar). In the first row, the first column is the classes of the model, and the last column is the *p* value (<0.05) which represents the significance of the classification model. The second and third column is the total number of samples in each class and their respective individual number of correct classifications. The last three rows indicate the correct classification of adulterated EBNs into non-authentic EBNs or incorrect classification into authentic EBNs class.

**A**	**Authentic**	**Non-authentic**	**Total**	***p* Value**
Quantity	98	8	106	4.87 × 10^−22^
Correct classification	100%	87.5%	99.1%	
1% adulteration		/		
5% adulteration		/		
10% adulteration	/	/		
**B**	**Authentic**	**Non-authentic**	**Total**	***p* value**
Quantity	98	10	108	1.34 × 10^−24^
Correct classification	100%	100%	100%	
1% adulteration		/		
5% adulteration		/		
10% adulteration		/		
**C**	**Authentic**	**Non-authentic**	**Total**	***p* value**
Quantity	98	100%	110	3.89 × 10^−16^
Correct classification	100%	75%	97.3%	
1% adulteration	/			
5% adulteration		/		
10% adulteration		/		
**D**	**Authentic**	**Non-authentic**	**Total**	***p* value**
Quantity	98	10	108	1.74 × 10^−10^
Correct classification	100%	40%	94.4%	
1% adulteration	/			
5% adulteration	/			
10% adulteration	/			
**E**	**Authentic**	**Non-authentic**	**Total**	***p* value**
Quantity	98	16	114	1.80 × 10^−28^
Correct classification	100%	87.5%	98.2%	
1% adulteration	/			
5% adulteration		/		
10% adulteration		/		

**Table 6 foods-11-02401-t006:** The summary of statistical sensitivity and specificity of VIS/SW-NIR and MIR techniques.

	VIS/SW-NIR		MIR	
Adulterants	Specificity	Sensitivity	Specificity	Sensitivity
Melamine	100%	75%	100%	87.5%
Karaya gum	100%	40%	100%	100%
Gelatine	100%	83.3%	100%	75%
Collagen	100%	100%	100%	40%
Nutrient agar	100%	100%	100%	87.5%

## Data Availability

The data that support the findings of this study are available from the corresponding author upon reasonable request.

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
