# Peer review of "Adulteration Detection of Edible Bird’s Nests Using Rapid Spectroscopic Techniques Coupled with Multi-Class Discriminant Analysis"

_foods, 2022, doi:10.3390/foods11162401_

Round 1
Reviewer 1 Report
This work is well designed and properly written. I believe it is interesting to publish it
Author Response
Thank you very much for your comments and suggestions.
Reviewer 2 Report
This is an interesting work and it provides useful information to the community of NIR spectroscopy and non-destructive methods of food analysis. At glance the study described in this manuscript looks nearly acceptable for being published. However, certain key elements of this manuscript require some attention. Importantly, several elements should be corrected or adjusted by the Authors. Most of those elements to be corrected are not resulting from the core experiment or the obtained results, but rather from the presentation of the study, background information and discussion.
The extent of the necessary changes is substantial, but at the same time the core of this study seems solid enough, resulting in a minor revision.
General remarks
1. It is important to highlight that this work does not really provide trustworthy conclusions on the capacity of NIR (and not even conventional NIR, but rather Vis/SW-NIR) vs. ATR-IR (or ATR-MIR) techniques to perform the analysis screened in this feasibility study. Therefore, I would tone down the claimed scope of providing comparative conclusions in this regard.
2. The NIR instrument used in this study is very peculiar, to say the least, as it is an ultra-miniaturized and ultra-cost-effective device primarily intended for consumer market. Moreover, it operates in a narrow spectral region of visible and short-wavelength NIR, is equipped with IrED light source and a simple Si array detector with just 12 independent photosensitive elements. SCiO sensors is very much different from most other popular miniaturized NIR spectrometers, and its application potential and performance is also somewhat specific. It is very important to provide the readership with clear information about the profound instrumental difference vs. more common NIR spectrometers and the resulting from technical principles of this sensor “instrumental difference”. Focused review literature would be helpful here (e.g. DOI: 10.1002/chem.202002838 and DOI: 10.1002/chem.202002838)
Clarity of presentation
3. The figures attempt to present rich information, however, with questionable success as most of those is unreadable, particularly score plots. I understand that it might not be easy to, but is necessary to provide readable information to the audience.
Design of experiment and methodology
4. It is dangerous and unadvisable to blindly consider and remove “outliers” in the data-set based solely on statistical tests. The presence of outliers is an important diagnostic information, and an attempt should always be made to identify the reason of specific samples to present themselves as outliers. Sample property and/or experimental conditions should be reviewed in each case independently, to gain understanding of the reason underlying the distribution of such spectra. However, no comments on this essential step can be found in the manuscript.
5. Pretreatment scheme for the spectral data was applied uniformly (i.e. to Vis/SW-NIR and ATR-IR), Moreover, just one pretreatment method was considered. Based on the rich literature evidence, most often the pretreatments need to be fine-tuned even for instruments operating in similar spectral region, let alone between Vis/SW-NIR and ATR-IR spectra, that are different in physical nature (e.g. by bandwidth, very broad bands in Vis/SW-NIR vs. sharp bands in MIR; this makes calculation of derivative as the pretreatment often very beneficial in Vis/SW-NIR). Therefore, this further demonstrates that this work did not provide a study exhaustive enough to draw more general conclusions about the suitability of “NIR vs MIR” (here I follow the manuscript’s convention, but see my previous comment regarding terminology) techniques to provide general conclusions. In other words, no attempt has been made to achieve optimal conditions for each analysis. Therefore, the positioning of this work presented by the Authors need to adjusted, to be fairer.
English/typing/style
- since MIR, NIR abbreviations are explained at the beginning of the manuscript, the further use of non-abbreviated terms (e.g. near-infrared) seems unnecessary;
- some minor polishing of the writing style should be carried out; however, in general the manuscript is easily comprehensible and easy to follow up to fine details;
In summary, this work is certainly publishable after a revision primary intended to improve the clarity and quality of presentation and remove potential points of misunderstandings.
Reviewer 3 Report
The Introduction sets the tone and context well and provides detailed and supported information.
Line 126: Is there a justification for the adulterants chosen to be investigated? Are these the most commonly detected in market and hence worth study? I suggest the authors provide added justification for better context to the reader.
Line 175: Clear and consistent use of units - Spectral range is reported in nm while Spectral resolution is reported in cm-1. Lack of consistent use of units is confusing. Suggest that the authors convert so all units are either nm or cm-1.
Line 206: Suggest changing from "most important information and reveals the hidden trend within the datasets" to "most important information allowing trends within the multivariate datasets to be observed." for readability.
Line 208: suggest changing from "without prior grouping of samples, but" to "without prior knowledge of sample grouping." for readability.
Line 209: suggest changing "their" to "known".
Line 210: The explanation here appears to contradict the previous statements on PCA by suggesting PCA is done by class. Could the authors provide more clarity on here by explaining whether PCA is being applied to each class as a separate investigation or to all the data as a whole?
Lines 212-215: The explanation of the cross-validation procedure is vague and confusing. Could the authors provide more detail on what "some part" of the data is, why differing numbers of PC's are used, how is prediction error assessed and what decides an optimum number of PC's.
Line 219: Suggest the authors only state indicators they use in the study, suggest removing the "and others" statement.
Line 220: Across the majority of statistical investigations the "normal" is for 5 or 10 fold cross validation. Why has 7 been used? Is this the software default?
Fig 3: The figure shows a large amount of spectral variation across the genuine EBN. Are the spectra in the figure displayed as the original measured spectra or the SNV processed? If there is this large amount of intensity variance have the authors considered what this may mean for application? Is a similar variance expected from the adulterants.
Line 271: suggest changing "identifying" to "approximating". The methods conveyed in the manuscript don't go as far as quantifying an exact LOD for each system but an indication of what can be determined as different from genuine based on different levels measured. At best the authors can determine a level at which they can observe an adulteration and at which the limit of detection will be known to be less than. Further investigation would be needed to comprehensively quantify a limit of detection for adulteration.
Line 281: Intensity of NIR peaks are dependent on a number of factors, in this particular case I would suggest the authors consider what the effects of pathlength and particle size would mean for the amount of light scattered back to the detector and how this would effect the spectral intensity - it may be that the sample morphology and method of analysis is having a greater effect on the change of intensity than is considered in the manuscript currently.
Fig 4: For greater understanding and context it is suggested that the authors also include the spectrum of the adulterant to each plot to provide a visual comparison.
Line 311: I don't believe shifted is the correct terminology, shifted implies a process or perturbation has occurred to change the molecular environment effecting the force constant and frequency of the vibration - hence shifting the observable wavelength of the peak. In this case the observation is on two different materials with different compositions - the peak is observed in different places for the different materials not shifted due to some change. The authors may consider rewriting this for clarity.
Fig 6: Suggest the same as for Fig 4.
Line 383: Suggest changing from "is" to "are".
Line 384: Suggest changing from "EBNs fall" to "EBNs which fall".
Line 384: Was the observation that outliers may be due to geographic origin investigated further. Suggest the authors provide more insight into whether the outliers come from one specific origin in particular or origins which are related in some way if possible.
Fig 7: How much variation is represented by each eigenvector (t) in these figures? Was an assessment made on how many eigenvectors contain the majority of information and how many contain a significant portion of the information? If not undertaken I would suggest the authors look into this further as t[3] may also contain useful information. Depending on the amount of variance represented by t[1] and t[2] the importance of t[1] may be far greater than t[2].
Line 424: suggest changing "regular" to "scores". A PCA scores plot does not have equal weight on each axis as does a regular scatter plot. The amount of variance represented by each eigenvector weights the interpretation of each axis, a common misconception of these plots is they read like a normal scatter plot and this can lead to misinterpretation. Clarity will help avoid this with the reader.
Line 427: suggest changing "overlapped" to "overlap". The use of suffixes on overlap isn't needed in the context of a number of sentences in the manuscript. Suggest the authors look at use of this through the manuscript.
Line 438: This is the first mention of "predictive score, t" in the manuscript even though it is used in the previous figures. It is suggested that the author explain t earlier in the manuscript for clarity to the reader. Potentially in the statistical methods section.
Line 455: suggest removing "Lastly". It does not add to the sentence.
Line 455: Is the misclassification assessment based on the k-folds cv method mentioned previously. Suggest the authors add clarity by explaining the origins of the data being discussed.
Table 4: The last 3 rows of the tables are unclear as to what they are conveying, it is suggested the authors provide more detailed explanation to clarify the information being conveyed.
Line 482: suggest changing from "was done by taking the" to "was undertaken using" to simplify the sentence.
Fig 11: Suggest same as for Fig 7.
Line 502: suggest removing the word "limit". For reasons previously discussed, the data does not quantify a limit but gives an indication of what was observed as different by the experiment.
Line 516: The initial part of the paragraph is repeating methodology explained earlier in the manuscript. Suggest this could be simplified by stating that "the same 2 class approach utilized for the NIR was applied to the MIR data".
Line 521: suggest changing "regular" to "scores" for reasons discussed previously.
Line 536: the information around interpreting the S-plot has been conveyed before in the manuscript. Suggest removing to avoid repetition.
Line 562: suggest changing "identify" to "investigate".
Line 565: It appears group and class are being used interchangeably throughout the manuscript. This can be confusing for readers. Suggest the authors review the use of the word "group" so it is always referring to the same concept to ensure a clear and consistent purpose for clarity to the reader.
Table 5: suggest the same as previously discussed for Table 4.
Line 595: There are different definitions for the word "sensitivity" depending on the analytical chemistry (change in response due to change in concentration) compared to statistics (number of genuine positive individuals predicted as positive). Suggest it is worth clarifying the intended definition used by the authors is the statistical.
Line 611: Suggest that it would also be worth analyzing and understanding the variation within the adulterant populations as it may effect application. The authors may wish to consider mentioning this also.
General suggestion: The clarity of the manuscript could be improved with some polishing of the sentence structure and English usage. The authors should consider having this further refined.
